# Traditional Uses, Pharmacological Activities, and Phytochemical Analysis of *Diospyros mespiliformis* Hochst. ex. A. DC (Ebenaceae): A Review

**DOI:** 10.3390/molecules28237759

**Published:** 2023-11-24

**Authors:** Thanyani Emelton Ramadwa, Stephen Meddows-Taylor

**Affiliations:** Department of Life and Consumer Sciences, College of Agriculture and Environmental Sciences, Florida Campus, University of South Africa, Private Bag X6, Florida 1710, South Africa; mtayls@unisa.ac.za

**Keywords:** traditional uses, pharmacology, phytochemicals, toxicity, *Diospyros mespiliformis*

## Abstract

*Diospyros mespiliformis* Hochst. ex. A. DC is widely distributed throughout Africa and around the world. It is utilized ethnobotanically to treat fevers, wounds, malaria, diabetes mellitus, and other diseases. This review aims to provide an exhaustive overview of the traditional uses, pharmacology, and phytochemical analysis of *D. mespiliformis*, with the objective of identifying its therapeutic potential for further research. Scientific resources, including Google Scholar, Science Direct, Web of Science, Pub Med, and Scopus, were used to find pertinent data on *D. mespiliformis*. Secondary metabolites tentatively identified from this species were primarily terpenoids, naphthoquinones, phenolics, and coumarins. *D. mespiliformis* has been reported to demonstrate pharmacological activities, including antimicrobial, antiproliferative, antiparasitic, antioxidant, anti-inflammatory, antiviral, anti-hypersensitivity, and antidiabetic properties. The phytochemicals and extracts from *D. mespiliformis* have been reported to have some pharmacological effects in in vivo studies and were not toxic to the animal models that were utilized. The *D. mespiliformis* information reported in this review provides researchers with a comprehensive summary of the current research status of this medicinal plant and a guide for further investigation.

## 1. Introduction

*Diospyros mespiliformis* Hochsr. ex. A. DC, commonly known as Jackal Berry or African Ebony and musuma in Venda, is a member of the Ebenaceae plant family. *Diospyros* is a general name that means “divine”, and *mespiliformis* comes from the Greek words “mesos”, which means “half”, and “pilos”, which means “bullets”. The plant is widely distributed throughout Africa, Asia, and parts of Europe [1]. In Africa, it can be found from as far as east Senegal, and all the way to Eritrea, Ethiopia, and Kenya, as well as in southern Namibia, northern South Africa, and Swaziland [2].

*D. mespiliformis* is a tall plant, with dense, rounded, and buttressed stems that can reach heights of 15 to 50 m, as shown in Figure 1. It grows in savannahs and woodlands, as well as along riverbanks in areas with regular rainfall, which support natural regeneration. The optimal conditions for the growth of the plant include a mean annual temperature of 16–27 °C, 500–1270 mm of annual rainfall, and an altitude of 350–1250 m [3]. The leaves are simple, alternately arranged, and dark green in color as presented in Figure 2. The plant is dioecious, flowers in April and May, and produces mature fruits in the form of big yellow berries [4]. Its bark is rough in texture and has a dense, evergreen canopy that is black to grey in color. The fruit is a fleshy berry with an enlarged calyx, which is yellow to orange when ripe [5].

Different plant parts have been well documented in ethnobotanical utilization to treat bacterial-, fungal-, viral-, parasitic-, and inflammatory-related ailments, among other things. The leaves are used as a treatment for fevers, as wound dressings, and as an antidote for a variety of poisonous substances. The roots and bark are used to treat diseases such as malaria, syphilis, and leprosy and to stop purging [6]. The bark and roots are also used traditionally for the treatment of diabetes mellitus in Vhembe district, Limpopo province, South Africa. Ground dry bark is mixed in hot water and, in some cases, mixed with *Bridelia micrantha* (Hochst) Baill bark, as well as *Elephantorrhiza elephantina* roots. The administered dosage is consumed through drinking a full cup twice daily [7]. *D. mespiliformis* has been reported to have a variety of pharmacological activities, including antimicrobial, antiproliferative, antiparasitic, antioxidant, anti-inflammatory, antiviral, anti-hypersensitivity, and antidiabetic properties. Preliminary phytochemical screening of *D. mespiliformis* revealed the presence of carbohydrates, flavonoids, saponins, tannins, phenols, steroids, triterpenoids, anthraquinones, anthocyanins, alkaloids, and cardenolides [8,9,10]. Different bioactive compounds have been tentatively identified in this medicinal plant species. Several studies have been undertaken to isolate bioactive compounds from the leaves, roots, stem, and bark. As a result, several secondary metabolites have been isolated, primarily from chemical constituents, such as terpenoids, naphthoquinones, phenolics, and coumarins [4,11,12,13,14]. No previous study has yet comprehensively reviewed the traditional uses, phytochemistry, pharmacological activities, and in vivo animal investigations of *D. mespiliformis*. Therefore, the current review aims to comprehensively examine and summarize the previously published articles regarding the traditional medicinal uses, phytochemistry, and pharmacological activities of this plant.

## 2. Results and Discussion

### 2.1. Traditional Uses

As shown in Table 1, different parts of *D. mespiliformis* are traditionally employed in the treatment of a wide range of disorders. Occasionally, various plant parts of the plant are utilized to treat the same conditions. Ringworm infections are treated topically using a decoction of the roots and leaves [15,16,17]. Furthermore, the bark is used in South Africa for ethnoveterinary purposes and for milk production [18]. The bark is boiled and the blend is ingested for stomach problems, such as to stop vomiting or diarrhoea, in Limpopo province, South Africa [19]. A root decoction is used to alleviate febrile symptoms by the Venda people of South Africa [20]. They crush the raw fruit, add a little water, and, thereafter, the infusion is used to treat fungal infections, particularly as a mouthwash or douche, 3× per day [21]. Decoctions of roots or leaves are used as infusions to treat urinary and sexually transmitted infections in Mpumalanga Province, South Africa [22]. In Burkina Faso, a decoction made from the bark and roots is used to alleviate toothache [23]. The roots are crushed and mixed with hot water, and the extraction is ingested to treat abdominal pain in the Midlands Province of Zimbabwe [24]. In Nigeria, a decoction of the root is usually ingested for a week or more to alleviate malaria [25]. In Zimbabwe, a log made of *D. mespiliformis* and *Gardenia spatulifolia* is placed at the base of the kraal for animals to leap over each day as a way to manage blackleg illnesses in cattle [26]. The fruits are pound and the juice is mixed with cow milk for drinking to treat dysentery in Benin [27]. A decoction obtained from the leaves or fruits, in association with the leaves of *Vitellaria parodoxa*, is given orally (0.5 L) and applied topically on the body for the treatment of skin diseases or tonic and zootechnically in veterinary medicine in northern Côte d’Ivoire, particularly for cattle [28]. The bark and roots are boiled in water, and the decoction is ingested to manage pneumonia and syphilis. The leaves are boiled in water and steaming is conducted to manage malaria fever. Pound leaves are rubbed into the skin for the treatment of skin diseases [29].

### 2.2. Phytochemical Analysis

Chivandi et al. [53] determined the lipid content and profile of oil from *D. mespiliformis* seeds. The total oil content of the seeds was low at 0.70 ± 0.17%. The lipid component of the seeds contained 39.54% saturated fatty acids and palmitic acid (C16:0), accounting for most of the saturated fatty acids at 30.06 ± 0.61%. The primary monounsaturated fatty acid was palmitoleic acid (C16:1n7), which accounted for 29.37 0.38% of the total monounsaturated fatty acids. Two polyunsaturated fatty acids were identified as linoleic acid (C18:2n6) and linolenic acid (C18:3n3), with levels of 28.71 ± 1.79% and 0.95 ± 0.61%, respectively. Hegazy et al. [54] determined the phytochemical content of the primary metabolites from *D. mespiliformis* fruits. The highest contents of total protein, hydrolysable carbohydrates, total soluble sugars, and free amino acids were found to be 9.28, 15.88, 9.82, and 2.95%, respectively. In another study by Ebbo et al. [55], the bark had the highest levels of vitamin E (140.91 ± 1.66 mg/dL), but low levels of vitamin A (1275 ± 2.90 mg/dL) and vitamin C (4.8 ± 0.11 mg/dL). The roots have the highest vitamin A concentration (1710 ± 577 mg/dL), as well as the highest levels of vitamin C (23.13 ± 0.43) and vitamin E (129.43 ± 0.42 mg/dL). The leaves had 1366.00 ± 6.88 mg/dL of vitamin A and 25.43 ± 1.18 mg/dL of vitamin C. Adewuyi et al. [56] evaluated the fatty acid composition and lipid profile of *D. mespiliformis* seed oils from Nigeria. The oils were analyzed for their fatty acid composition, lipid classes, distribution of fatty acids in the lipid fractions, and molecular speciation of the phospholipids, glycolipids, and triacylglycerols. *D. mespiliformis* was found to have an oil yield of 4.72 ± 0.2%. It was discovered that *D. mespiliformis* contained 0.84 ± 0.10 g/100 g of C12:0 and 0.82 ± 0.10 g/100 g of C14:0 fatty acids, respectively. Additionally, C18:2 was shown to be the most prevalent fatty acid (34.97 ± 0.40 g/100 g fatty acids). Exactly 60.14 g of unsaturated fatty acids was detected. The outcomes were in line with earlier research on *D. mespiliformis* fatty acid levels conducted by Chivandi et al. [53]. The neutral lipid content was 93.60 ± 0.20%. C16:1 was exclusively found in the neutral (0.41 ± 0.05 g/100 g fatty acids) and glycolipid (0.36 ± 0.05 g/100 g fatty acids) lipids, but not in the phospholipids.

Petzke et al. [57] conducted research on *D. mespiliformis* seeds to ascertain their nitrogen and amino acid contents, chemical score, protein-digestibility-corrected amino acid score, accessible lysine, and in vitro digestibility. The seeds contained 5.44% crude nitrogen, 0.87% nitrogen, and 8.99% moisture. Moreover, arginine (501 mg/gn), aspartic acid (507 mg/gn), and glutamic acid (1002 mg/gn) were found to be more abundant in *D. mespiliformis* seeds than other amino acids. Cysteine and methionine (95%) and tryptophan (75%), respectively, had the highest percentages when it came to the protein-digestibility-corrected amino acid score of important amino acids and in vitro protein digestibility in *D. mespiliformis* seed samples. The amino acid, fatty acid, and mineral contents of yari, a mixture of lichens that primarily consists of *Rimelia reticulate* and grows on *D. mespiliformis*, were examined by Glew et al. [58] in their analysis of plant food in West Africa. When the separate amino acid contents were added together, the estimated protein concentration of *D. mespiliformis* was 5.31% in yari. The food’s proportions of necessary amino acids did not surpass that of the ideal protein, lysine, in yari (74%). The fatty acid compositions of the plant food are expressed on a dry weight basis. The plant contained less than 1% fatty acid of yari (0.25%). More than 15 mg/g of dry weight calcium was found exclusively in yari, a plant meal used as a condiment. The dry weight of *D. mespiliformis* exhibited a zinc concentration ranging from 12.1 to 19.0 µg. Achaglinkame et al. [59] determined the nutritional characteristics of wild fruits from *D. mespiliformis* that are of dietary interest in Ghana. According to the proximate and physicochemical characteristics of the fruits, *D. mespiliformis* had the maximum moisture content of 6%, but in dry matter, its highest percentage was 93.99%. The fruits’ ash content was 3%, the crude fiber content was 2%, and the pH value was 5.44, meaning that the fruits are quite acidic. The fruits of *D. mespiliformis* exhibited the highest levels of magnesium (162.98 ± 0.42), potassium (129.4 ± 1.62), and phosphorus (64.78 ± 2.98) in terms of mineral content (mg/100 g dry weight). The fruits of *D. mespiliformis* had the highest vitamin composition (mg/100 g) of vitamin B3 (310.22 ± 8.15).

### 2.3. Secondary Metabolites

Maitera et al. [10] evaluated the tannin content accumulated in the unripe fruit, leaves, and bark of *D. mespiliformis* extracted with acetone, methanol, 70% methanol, and hot- and cold-water extracts. According to the study, unripe fruits had the most tannin content, but the weight of the extracts from 100 g of the powdered material in 70% methanol (15.94 g) and acetone (13.52 g) was much greater. Furthermore, the weight of the leaves after tannin extraction in acetone and 70% methanol extracts was 12.35 g and 11.55 g, respectively, while the weight of the bark after tannin extraction in acetone was 12.33 g. The root bark aqueous extract of *D. mespiliformis* was quantitatively analyzed by Vandi et al. [8] for the presence of various phytochemicals, including polyphenols (86.58), flavonoids (55.22), tannins (21.71), anthocyanins (10.14), and saponins (21.92) (mEq/100 g of dry).

### 2.4. Isolated or Tentatively Identified Compounds from D. mespiliformis

According to multiple studies on phytochemistry analyses using various chromatographic and spectroscopic techniques, there are triterpenes such as *α*-amyrin-baurenol (**21**), trihydroxy-triterpenoid acid (**32**), *α*-amyrin (**19**), *β*-sitosterol (**29**), lupeol (**27**), betulin (**24),** and betulinic acid (**25**) in the stem bark and wood of *D. mespiliformis* [41,60,61] in addition to naphthoquinones, e.g., diospyrin (**1**), isodiospyrin (**3**), diosquinone (**2**), and plumbagin (**12**) [62,63], as shown in Table 2. Additionally, Mohamed et al. [14] isolated lupeol (**27**), betulin (**24**), betulinic acid (**25**), and lupenone (**26**) from the stems and bark of *D. mespiliformis*. Anas et al. [13] and Adzu et al. [61] have reported the isolation and identification of lupeol (**27**) from the stem bark of *D. mespiliformis*. Diosquinone (**2**) and plumbagin (**12**) were also isolated from the roots of *D. mespiliformis* by Lajubutu et al. [64].

Ultra-performance liquid chromatography-electrospray ionization-mass spectrometry was used to tentatively identify several secondary metabolites from the methanol extract of *D. mespiliformis*, comprising kaempferol (**5**), myricetin (**11**), quercetin (**13**), 4,4′,6,7-tetrahydroxyaurone (**17**), 8-methoxy-3-methyl-1,2-naphthoquinone (**10**), and the tetrahydrodiospyrin (**40**), as shown in Figure 3. Three lupane-type triterpenes (30-hydroxylup-20(29)-en-3*β*-ol, betulinaldehyde (**23**), betulinic acid (**25**)) and betulafolienetriol (**22**) were also identified alongside *δ*-tocopherol (**18**) and the pentagallic acid ester of glucose (**39**), as shown in Figure 4 [12]. In another study, Dangoggo et al. [4] tentatively identified secondary metabolites from *D. mespiliformis* leaves using Fourier transform infrared spectroscopy and gas chromatography-mass spectroscopy (GC-MS). Three compounds were identified, namely, 4-hydroxyl-4-methylpentan-2-one (**34**), octadecanoic acid (**35**), and 1-octadecyne (**37**), as presented in Figure 5. David et al. [11] recently conducted GC–MS analysis of dichloromethane fractions from a woody stem methanol extract and highlighted the presence of natural products such as pentadecanoic acid (**35**), octadecanoic acid methyl ester (**36**), *cis*-vaccenic acid (**41**), *β*-sitosterol (**19**), lupeol (**27**), stigmastan,3,5-diene (**31**), and a lupeol derivative: 3*β*-lup-20(30)-en-3-olacetate (**28**). Following an investigation of *D. mespiliformis* leaves by Hawas et al. [65], a new acylated flavone isoscutellarein 7-*O*-(4′′′-*O*-acetyl)-*β*-allopyranosyl (1′′′ → 2″)-*β*-glucopyranoside (**4**) was isolated and characterized. Furthermore, eight known flavonoid metabolites were identified: luteolin 3′,4′,6,8-tetramethyl ether (**9**), luteolin 4′-*O*-*β*-neohesperidoside (**10**), luteolin 7-*O*-*β*-glucoside (**7**), luteolin (**6**), quercetin (**13**), quercetin 3-*O*-*β*-glucoside (**14**), quercetin 3-*O*-*α*-rhamnoside (**15**), and rutin (**16**). In addition, their structures were determined via acid hydrolysis of the separated glycosides and via spectroscopic (UV, NMR, and MS) data analyses.

**Table 2 molecules-28-07759-t002:** Reported isolated or tentatively identified compounds from *D. mespiliformis*.

No.	Compounds	Plant Part	Detection/Isolation Method	Reference
**1**	Diospyrin	Stem bark or wood	Isolated	[61,62]
**2**	Diosquinone	Stem bark, wood, roots	Isolated	[60,61,62]
**3**	Isodiospyrin	Stem bark or wood	Isolated	[60,61]
**4**	Isoscutellarein 7-*O*-(4′′′-*O*-acetyl)-*β*-allopyranosyl (1′′′→2″)-*β*-glucopyranoside	Leaves	Isolated	[66]
**5**	Kaempferol	Stem bark	UPLC-ESI-MS	[12]
**6**	Luteolin	Leaves	Isolated	[66]
**7**	Luteolin 7-*O*-*β*-glucoside	Leaves	Isolated	[66]
**8**	Luteolin 4′-*O*-*β*-neohesperidoside	Leaves	Isolated	[66]
**9**	Luteolin 3′,4′,6,8-tetramethyl ether	Leaves	Isolated	[66]
**10**	8-methoxy-3-methyl-1,2-naphthoquinone	Stem bark	UPLC-ESI-MS	[12]
**11**	Myricetin	Stem bark	UPLC-ESI-MS	[12]
**12**	Plumbagin	Stem bark, wood, roots	Isolated	[60,61,65]
**13**	Quercetin	Stem bark, leaves	Isolated, UPLC-ESI-MS	[12,66]
**14**	Quercetin 3-*O*-*β*-glucoside	Leaves	Isolated	[66]
**15**	Quercetin 3-*O*-*α*-rhamnoside	Leaves	Isolated	[66]
**16**	Rutin	Leaves	Isolated	[66]
**17**	4,4′,6,7-Tetrahydroxyaurone	Stem bark	UPLC-ESI-MS	[12]
**18**	*δ*-Tocopherol	Stem bark	UPLC-ESI-MS	[12]
**19**	*β*-Amyrin	Seeds	GC-MS	[54]
**20**	*α*-Amyrin	Stem bark or wood	Isolated	[41,60,61,63]
**21**	*α*-Amyrin-baurenol	Stem bark or wood	Isolated	[41,60,61,63]
**22**	Betulafolienetriol	Stem bark	UPLC-ESI-MS	[12]
**23**	30-Hydroxylup-20(29)-en-3*β*-ol, betulinaldehyde	Stem bark	UPLC-ESI-MS	[12]
**24**	Betulin	Stem bark or wood	Isolated, GC-MS	[14,41,60,61,63]
**25**	Betulinic acid	Stem bark or wood	Isolated, UPLC-ESI-MS	[12,14,41,60,61,63]
**26**	Lupenone	Stem bark	Isolated	[14]
**27**	Lupeol	Stem bark or wood	Isolated, GC-MS	[11,13,14,41,60,61,63,64]
**28**	3*β*-Lup-20(30)-en-3-olacetate	Wood stem	GC-MS	[11]
**29**	*β*-Sitosterol	Stem bark or wood	Isolated	[11,41,60,61,63]
**30**	*γ*-Sitosterol	Seeds	GC-MS	[54]
**31**	Stigmastan,3,5-diene	Wood stem	GC-MS	[11]
**32**	Trihydroxy-triterpenoid acid	Stem bark or wood	Isolated	[41,60,61,63]
**33**	Hexadecane	Seeds	GC-MS	[54]
**34**	4-Hydroxyl-4-methylpentan-2-one	Leaves	GC-MS	[4]
**35**	Octadecanoic acid	Leaves, wood stem	GC-MS	[4,11]
**36**	Octadecanoic acid methyl ester	Wood stem	GC-MS	[11]
**37**	1-Octadecyne	Leaves	GC-MS	[4]
**38**	Octadiene	Seeds	GC-MS	[54]
**39**	Pentagallic acid ester of glucose	Stem bark	UPLC-ESI-MS	[12]
**40**	Tetrahydrodiospyrin	Stem bark	UPLC-ESI-MS	[12]
**41**	*cis*-Vaccenic acid	Wood stem	GC-MS	[11]

### 2.5. Pharmacological Activity

A summary of the pharmacological activities of the different parts and major compounds from *D. mespiliformis* is described in Table 3.

#### 2.5.1. Antimicrobial Activity

Esimone et al. [5] tested the leaf and root extracts of *D. mespiliformis* in methanol and water, as well as their combination, for possible antimycobacterial activity against *Mycobacterium smegmatis*. The methanol leaf and root extracts of *D. mespiliformis* had minimum inhibitory concentrations (MICs) of 167 µg/mL and 250 µg/mL, respectively. The highest synergistic antimycobacterial activity was shown by the 8:2 ratio of *D. mespiliformis* and *Anthocleista djalonensis* against *M. smegmatis* [5]. Green et al. [50] used tetrazolium microplate tests to determine the MIC of *D. mespiliformis* hexane leaf extracts against *Mycobacterium tuberculosis* H_37_R_a_, a clinical strain that was resistant to first-line drugs and one second-line drug. The MIC of the hexane leaf extracts against both *M. tuberculosis* H_37_R_a_ and the clinical isolate was 100 µg/mL.

The antibacterial activity of ethanol extracts of *D. mespiliformis* was assessed by Dangoggo et al. [4] using the disc diffusion method. The ethanol leaf extracts of *D. Mespiliformis* inhibited *E. coli* at concentrations of 90 mg/mL and 120 mg/mL, and *P. aeruginosa* was inhibited at concentrations of 12 mg/mL and 13 mg/mL. The zone of inhibition for the water extract was 10–13 mm on *S. aureus* at 30–90 mg/mL and 120 mg/mL, 11–13 mm on *P. aeruginosa* and 11–14 mm on *E. coli* at 90–120 mg/mL, and 10–11 mm at 90–120 mg/mL and 120 mg/mL on *Shigella* spp. *D. mespiliformis* dichloromethane and methanol crude extracts have been investigated for their antibacterial properties by Mabona et al. [15]. The MIC values of the extracted mixture of dichloromethane and methanol against *Propionibacterium acnes* ATCC 11827 and *Trichophyton mentagrophytes* ATCC 9533 were 50 µg/mL and 100 µg/mL, respectively. Shai et al. [66] determined the antibacterial activity of acetone leaf extracts against twenty different bacterial species and *D. mespiliformis* acetone leaf extracts had an MIC of 80 µg/mL against *Bacillus stearothermophilus*.

According to a study by Shikwambana and Mahlo [86], the aqueous leaf and bark extracts of *D. mespiliformis* had excellent antifungal activity against *Candida albicans* with an MIC of 20 µg/mL and good activity against acetone leaf extracts with an MIC of 80 µg/mL after 48 h. Additionally, *D. mespiliformis* acetone leaf extracts demonstrated strong antifungal activity against *Microsporum canis* with an MIC value of 40 µg/mL after 48 h and outstanding antifungal activity against *M. canis* with MIC values of 20 µg/mL after 24 and 48 h. With regards to *Trichophyton rubrum*, the aqueous and acetone leaf extracts showed outstanding activity, with MICs of 20 µg/mL after 24 and 48 h. Previous investigations indicated that acetone extracts had an excellent antifungal activity [83]. According to the study conducted by Mamba et al. [33], the MICs of *D. mespiliformis* 70% ethanol leaf extracts ranged between 3.1 and 6.3 mg/mL against *C. albicans* ATCC 10231, *Gardnerella vaginalis* ATCC 14018, *Neisseria gonorrhoeae* ATCC 19424, and *Olivella ureolytica* ATCC 43534. Hawas et al. [65] tested the antimicrobial activity of the flavonoids that were isolated from *D. mespiliformis* leaves against four human pathogenic bacteria. Flavonol *O*-rhamnoside (**15**) had moderate activity against *S. aureus*, with an MIC value of 9.77 μg/mL, while methylated flavone showed strong action against *E. coli*, with an inhibition zone of 34 mm. The minimal bactericidal concertation (MBC)/MIC ratio was used to assess the antibacterial activity of the isolated flavonoids. Furthermore, the study discovered that flavonoids were bactericidal against *S. aureus* and that flavonoids were bactericidal against *E. coli*. Lajubutu et al. [64] tested the antibacterial activity of diosquinone (**2**) and plumbagin (**12**) that were isolated from the roots of *D. mespiliformis* [64]. The diosquinone (**2**) MICs ranged from 3 to 30 μg/mL for *S. aureus* NCTC 6571 and *S. aureus* E3T, while they were 15 to 16 μg/mL for *E. coli* KL16 and *P. aeruginosa* NCTC 6750. Furthermore, *S. aureus* NCTC 6571 responded paradoxically and biphasically to diosquinone (**2**) in nutritional broth, yet its bacterial activity against *E. coli* KL16 increased as the concentration rose to the maximum diosquinone (**2**) concentration measured.

#### 2.5.2. Anti-Inflammatory Activity

Adzu et al. [61] evaluated the in vivo antipyretic, analgesic, and anti-inflammatory effects of *D. mespiliformis* methanol stem bark extracts in rats and mice. The extract demonstrated significant efficacy (PB/0.05) against all analgesic and anti-inflammatory models applied at 100 mg/kg and had an antipyretic effect at 50 and 100 mg/kg i.p. According to the findings, the extract’s LD_50_ in mice was 513.809 ± 33.92 mg/kg i.p. In a different investigation, Adzu et al. [80] extracted *D. mespiliformis* stem bark progressively using hexane, chloroform, and methanol, and then performed preliminary analgesic action on the extract. The most active of the three extracts was the chloroform extract, which was also subjected to column chromatography, resulting in the isolation of lupeol (**27**). In rats, lupeol (**27**) reduced the pain stimulus brought on by the analgesic meter and formalin. It has been found that lupeol (**27**) functions either individually or together with different compounds, and it may have been responsible for the plant’s benefits in the treatment of pain-related disorders.

Mamba et al. [33] evaluated the anti-inflammatory properties of *D. mespiliformis* root extracts using the 15-lipoxygenase (15-LOX) model of inhibition. The IC_50_ value for the *D. mespiliformis* root extract’s anti-inflammatory effects was only 188.1 µg/mL. In a different study, Lawal et al. [72] revealed that *D. mespiliformis* had a modest activity against xanthine oxidase (XO), with an IC_50_ value of 142 8 µg/mL, but had an inhibitory effect against 15-LOX at the highest tested dose. At the maximum measured concentration of 100 µg/mL, *D. mespiliformis* extracts suppressed the formation of nitric oxide (NO) by around 68.1%, which is just under the 70% threshold. The immunomodulatory effects of solvent fractions of *D. mespiliformis* were investigated by David et al. [11] in mice infected with a *Plasmodium berghei* (NK 65)-sensitive strain. Compared to the pharmacological control, the levels of IgG, IgM, and tumor necrosis factor alpha (TNF) were considerably greater in the dichloromethane fraction group, although the interleukin 1 beta (IL-1β) and interleukin 6 (IL-6) values did not change proportionally with the dose. The study established that the dichloromethane fraction had immunomodulatory effects on infected mice.

Ebbo et al. [55] highlighted the *D. mespiliformis* crude bark, leaf, and root methanol extracts’ wound-healing capabilities using in vivo animal models. After 11 days of treatment with the crude methanol extracts of the bark and roots of *D. mespiliformis*, the rats’ dorso-caudal lesions were healed. Exactly 13 days after the initial wound, the wound was healed in the same amount of time as rats given penicillin and a leaf extract of *D. mespiliformis*. The lesions on the rats treated with carboxyl methyl cellulose were healed exactly 15 days after it was administered. Swelling and reddening were noticeable in the bark, leaf, and root treatment groups throughout the first five days of the trial. On day 9, the wounds of rats fed with *D. mespiliformis* bark fractions in ethyl acetate and hexane had fully healed. Animals given butanol and water fractions did not statistically differ (*p* > 0.05) from groups given ethyl acetate and hexane extracts. On the eleventh day following wounding, complete wound closure was attained in all groups for the *D. mespiliformis* leaves. On day 9, the water-fraction-treated group had the largest incision length, measuring 3.5 ± 0.29 mm. This was statistically (*p* < 0.05) greater than the incision lengths for the hexane- and ethyl-acetate-treated groups, which were 1.0 mm on the experiment day. In the same experiment, Ebbo et al. [55] investigated the in vitro anti-inflammatory activity of various *D. mespiliformis* fractions against the LOX-15 enzyme. The hexane fraction showed the maximum percentage of inhibition at 10 mg/mL and 5 mg/mL, with values of 32.05 ± 2.79 and 31.21 ± 0.84, respectively. The water component was inhibited by 19.67 ± 2.29 percent at a dosage of 10 mg/mL compared to zero at a concentration of 5 mg/mL. The butanol and ethyl acetate fractions at 5 and 10 mg/mL appear to activate the enzyme.

#### 2.5.3. Antiparasitic Activity

Aderbauer et al. [67] examined the in vitro antitrypanosomal activity of dichloromethane leaf extracts from *D. mespiliformis* against *Trypanosoma brucei* in a long-term viability assay. The crude extract had a poor MIC of only 500 µg/mL against *T. brucei*, which is higher than the criterion for pharmacological significance of 100 µg/mL. Nafuka [73] evaluated the in vitro antiplasmodial effectiveness of methanol and aqueous extracts of *D. mespiliformis* (leaf and root) against *Plasmodium falciparum*. After *P. falciparum* 3D7A had been treated with crude methanol leaf extracts from *D. mespiliformis* for 24 and 48 h, the average percentage of parasitemia decreased across all concentrations; however, this was not statistically significant (*p* = 0.3 and 0.5, respectively). There was only time-dependent antiplasmodial activity at 5 µg/mL and 24 h efficacy at 10 µg/mL. The average percentage of parasitemia for the aqueous extract did not decrease statistically significantly at 24 h (*p* = 0.6) or 48 h (*p* = 0.1). At 24 h as opposed to 48 h, the leaf extract was more effective against *P. falciparum* 3D7A. At 24 h, the leaf extract was more effective against *P. falciparum* 3D7A than at 48 h. *D. mespiliformis* aqueous root extracts had an IC_50_ of 2.91 µg/mL and leaf extracts had an IC_50_ of 3.01 µg/mL for aqueous extracts. The *D. mespiliformis* methanol leaf extracts were the most effective, with an IC_50_ of 1.51 µg/mL, while the methanol root extracts also had good activity at 2.12 µg/mL [86].

The in vitro antiplasmodial activity of root extracts from *D. mespiliformis* was evaluated by Bapela et al. [20]. The extractant utilized on the powdered root material was dichloromethane: 50% methanol (1:1). Antiplasmodial activity against the chloroquine-sensitive strain of *Plasmodium falciparum* (NF54) was investigated. Polar extracts from the roots of *D. mespiliformis* prevented the growth of plasmodial cells (IC_50_ = 28.4 µg/mL) and showed significant dichloromethane action (IC_50_ = 4.40 µg/mL). The in vitro antiprotozoal activity of *D. mespiliformis* leaf extracts in 70% ethanol was discovered by Traore et al. [69]. The *Trypanosoma brucei brucei* IC_50_ values for *D. mespiliformis* were 25.8 µg/mL for *Trypanosoma cruzi*, >64 µg/mL for *Leishmania infantum*, and 24.9 µg/mL for *Plasmodium falciparum*. The effect of a crude ethanolic extract of *D. mespiliformis* on the clinicopathological variables of Yankasa sheep experimentally infected with *Haemonchus contortus* was determined by Luka et al. [25]. Throughout the duration of the trial, L3 *H. contortus* larvae infection of Yankasa sheep did not result in statistically significant alterations (*p* > 0.05) in the mean rectal temperature. The extract showed some effectiveness against *H. contortus* at the tested dosages. Bapela et al. [76] evaluated the inhibitory effects of *D. mespiliformis* dichloromethane (DCM) and 50% MeOH root extracts against axenically grown amastigote forms of *Leishmania donovani* (MHOM-ET-67/L82). A considerable antileishmanial effect was shown by the *D. mespiliformis* DCM root extract, with an IC_50_ of 7.7 µg/mL, while some antileishmanial effects were observed for 50% MeOH root extracts, with an IC_50_ of 54 µg/mL.

The in vivo antimalarial effects of bark extracts of *D. mespiliformis* were examined by Chinwe et al. [81] in adult Swiss albino mice that had been infected with a chloroquine-resistant NK65 lineage of *Plasmodium berghei*. Bark extracts of *D. mespiliformis* demonstrated a more substantial antimalaria efficacy, with an inhibition percentage of 53% at a dose of 800 mg/kg. Agbadoronye et al. [77] investigated the antitrypanosomal activities of *D. mespiliformis* leaf extracts and an alkaloidal fraction in *Trypanosoma evansi*-infected rats. White blood cells, the packed cell volume, the mean corpuscular hemoglobin, the mean corpuscular hemoglobin concentration, and crude extracts at 400 mg/kg BW and 100 and 200 mg/kg BW substantially (*p* < 0.05) increased the red blood cells and elevated bilirubin levels while decreasing the crude extract. Furthermore, the extract considerably reduced the total proteins. The effectiveness of *D. mespiliformis* against chloroquine-sensitive and -resistant strains of malarial parasites in mice was examined by Olanlokun et al. [12]. At 400 mg/kg, *D. mespiliformis* decreased mean percentage parasitemia values (5 ± 1), increased the packed cell volume (36% ± 1.4), and increased platelets (2 ± 1.4). At the same dose, *D. mespiliformis* decreased the activities of alkaline phosphatase (56 ± 0.7 U/L), alanine aminotransferases (6.2 ± 0.8 U/L), and alanine aminotransferases (8 ± 3.8 U/L). *D. mespiliformis* reversed the start of the permeability transition while decreasing ATPase enhancement and lipid peroxidation. Although *D. mespiliformis* was effectively tolerated at the maximal dose in the infected control group, liver histology in that group showed severe widespread congestion and wide hemorrhagic lesions. In an additional investigation carried out by Olanlokun et al. [78], the antiplasmodial effects of *D. mespiliformis* root extracts were investigated in *Plasmodium berghei*-infected mice. According to the results, *D. mespiliformis* had a high rate of parasite clearance (84.7%) and a lower proportion of parasitemia (0.67%). The fractions and extracts of *D. mespiliformis* considerably reduced the production of β-hematin due to their cell-free antiplasmodial activity. David et al. [11] carried out a study to evaluate the bioactivity-guided antiplasmodial efficacy of solvent fractions of *D. mespiliformis* in mice infected with a susceptible strain of *Plasmodium berghei* (NK 65). The crude methanol extract of the stems of *D. mespiliformis* was partitioned between *n*-hexane, dichloromethane, ethyl acetate, and methanol. The dichloromethane fraction had the highest parasite clearance and improved hematological indices relative to the drug control. The heme values increased, while the hemozoin content significantly (*p* < 0.05) decreased. The highest dose of *n*-hexane and methanol opened the mitochondrial permeability transition (mPT) pore, while the reversal effects of dichloromethane on the mPT, mitochondrial F_1_F_0_ ATPase, and lipid peroxidation were dose-dependent.

#### 2.5.4. Antidiabetic Activity

Mohamed et al. [14] studied the α-glucosidase enzyme inhibition activity of isolated bioactive compounds from *D. mespiliformis*. Lupeol, botulin, and lupenone had an α-glucosidase inhibitory activity, with an IC_50_ ranging from 0.002 to 0.46 mM.

#### 2.5.5. Antiviral Activity

*D. mespiliformis* root extracts were tested for anti-HIV activity against recombinant HIV-1 enzyme by Mamba et al. [33] using a non-radioactive HIV-RT colorimetric assay. With a 17.4% inhibition of the HIV-1 RT, the root extracts of *D. mespiliformis* demonstrated a poor inhibitory efficacy. Similar to the findings of Hedimbi [68], who demonstrated that *D. mespiliformis* leaf extracts at 0.1 mg/mL had 78.7% HIV-1 RT activity, it was confirmed that extracts of *D. mespiliformis* have varying degrees of activity against HIV-1 RT. Chukwuma [70] investigated the antiviral activities of *D. mespiliformis* aqueous, ethanolic, and methanolic extracts on the avian viruses Newcastle disease virus (NDV), fowl pox virus (FPV), and infectious bursal disease virus (IBDV). Aqueous extracts of *D. mespiliformis* at 400 mg/mL, 200 mg/mL, and 100 mg/mL, respectively, inhibited the virus (NDV) in percentages of 91%, 86%, and 85%. The percentage inhibition of the ethanolic extracts was 95%, 90.5% and 89%, respectively. At a concentration of 400 mg/mL of the crude extract of *D. mespiliformis*, the tested FPV showed an extremely high activity. All plant extracts were shown to have 100% egg mortality at the end of the experiment with the infectious bursal disease virus (IBDV).

#### 2.5.6. Anti-Hypersensitivity

Belemtougri et al. [71] tested the efficacy of the crude, aqueous, and ethanolic extracts of *D. mespiliformis* to inhibit the effects of caffeine on the release of calcium from the sarcoplasmic reticulum of rat skeletal muscle cells. Different *D. mespiliformis* extracts failed to function in rat skeletal muscle cells when applied alone, demonstrating that they are unable to change the resting calcium levels of skeletal muscle cells. When caffeine (10 mmol/L) was given to myotubes, Ca^2+^ was released from the SR. The reaction was used as a control, and each cell was given a 10 mmol/L caffeine solution. Then, utilizing caffeine and plant extracts, a second cell was investigated. The crude extract of *D. mespiliformis* at a concentration of 10 mg/mL reduced the amplitude of Ca^2+^ release from the SR. This proved that these extracts significantly restrict the release of Ca^2+^, which is sensitive to caffeine from the SR. The suppression of intracellular calcium release by various extracts was dose-dependent, with crude decoctions being the most effective. The following are some categories for the effects of several *D. mespiliformis* extracts at 10 mg/mL: The aqueous extract follows the ethanolic extract after the crude decoction. The crude decoction contains 51% *D. mespiliformis*, with an IC_50_ of 8.84 mg/mL at the same dose. However, the IC_50_ of the *D. mespiliformis* ethanolic extract was 9.23 mg/mL, whereas the IC_50_ of the other extracts exceeded 10 mg/mL. Calcium release from the sarcoplasmic reticulum was inhibited in ethanolic extracts by 54% and in aqueous extracts by 29% in 10 mg/mL crude *D. mespiliformis* decoctions.

#### 2.5.7. Antioxidant Activity

Ndhlala et al. [85] investigated the methanol extracts of *D. mespiliformis* wild fruits and analyzed them for their scavenging effect of the 1,1-diphenyl-2-picrylhydrazyl (DPPH) radical, reducing power, and anion radical effect on superoxide anions using a colorimetric method. There was an increase in the radical-scavenging effect, reducing power, and superoxide-anion-radical-scavenging effect as the concentration of the sample increased. *D. mespiliformis* had a high DPPH-radical-scavenging capacity. According to Sombie et al. [87], the leaf extracts of *D. mespiliformis* exhibited a TEAC of 1.170 ± 00 mM (ABTS) (*p* < 0.05), which was a significantly higher antioxidant capacity than Trolox (1 mM). Extracts of *D. mespiliformis* (1.17 ± 0.00 mM TEAC/g and 70.77 ± 0.4 M ET/g) showed the greatest antioxidant activity. The acetone leaf extracts of *D. mespiliformis* were also tested for their radical scavenging potential and had an IC_50_ of 25 ± 2 μg/mL in a DPPH assay [80]. *D. mespiliformis* fruit extracts were studied for their ability to scavenge DPPH by Hegazy et al. [54]. *D. mespiliformis* displayed a greater level of DPPH-radical-scavenging activity (87.36%) at a concentration of 1 mg/mL of the radical. A higher hydrogen peroxide scavenging activity than 85% inhibition was demonstrated by the *D. mespiliformis* extracts at methanol concentrations of 1 mg/mL. The antioxidant capacity of the powdered *D. mespiliformis* fruit was investigated in relation to the effects of solvent extractions (ethanolic and hydroethanolic extracts) [84]. The IC_50_ values of DPPH-radical-scavenging activity were found to be 1.037 ± 0.204 mg/mL for ethanolic extracts and 1.111 ± 0.133 mg/mL for hydroethanolic extracts. To evaluate the in vivo antioxidant activity against high-fat diet (HFD)-induced hyperlipidemia in rats and different particle size powder fractions, ethanolic and hydroethanolic extracts of *D. mespiliformis* fruits were administered orally (600 mg/kg, p.o.) for 30 days with a HFD, and the effect of the extracts on enzymatic antioxidants like superoxide dismutase (SOD), catalase (CAT), and peroxidase was estimated in the blood, heart, liver, and kidneys [78]. In comparison to the control group, various samples of *D. mespiliformis* fruit powder considerably increased the levels of SOD, catalase, peroxidase, alanine transaminase, and aspartate aminotransferase enzymes.

In vitro antioxidant activities of root bark aqueous extracts of *D. mespiliformis* (ABTS, DPPH and FRAP) were determined [8]. The ABTS radical’s inhibitory concentration of 50% (IC_50_) was 220 µg/mL, and the root bark aqueous extract had the ability to scavenge ABTS and DPPH radicals as well as reduce FRAP. Ebbo et al. [55] showed the DPPH-radical-scavenging properties of crude methanol extracts and fractions of *D. mespiliformis* leaves, bark, and roots. The crude methanol extracts of the leaves, bark, and roots of *D. mespiliformis* had IC_50_ values of 6.94 ± 0.49 µg/mL, 7.82 ± 0.76 µg/mL, and 3.47 ± 0.05 µg/mL, respectively. The ethyl acetate fraction showed the lowest IC_50_ (1.08 ± 0.04 µg/mL) and the greatest antioxidant activity. Antioxidant activity was observed in both the water and butanol fractions, with IC_50_ values of 4.73 µg/mL and 1.44 µg/mL, respectively. Hawas et al. [65] evaluated the antioxidant activity of *D. mespiliformis* secondary metabolites using a DPPH radical-scavenging assay. The new acylated flavone (**8**) and flavonol *O*-rhamnoside (**15**) demonstrated the most potent antioxidant activity, with IC_50_ values of 15.46 μg/mL and 12.32 μg/mL, respectively.

#### 2.5.8. Antiproliferative Activity

Adeniyi et al. [36] assessed the cytotoxicity activity of diosquinone (**17**) previously isolated from the root bark of *D*. *mespiliformis* against ten cancer cell lines (human breast (BC-1), colon (COL-2), human fibrosarcoma (HT-1080), human lung cancer (LU-1), human nasopharyngeal carcinoma (KB), oral epidermoid carcinoma (KB) and KBV1, prostrate (LNCaP), human glioblastoma cells (U373), human neuroblastoma (SKNSH), multiple-drug-resistant or vinblastine-resistant human nasopharyngeal carcinoma (KB-V(V-VLB))). Diosquinone (**2**) was more active against human glioblastoma with an ED_50_ of 0.18 μg/mL. It is interesting to note that naphthoquinone epoxide significantly inhibits BC-1, HT-1080, Lu-1, KB, and SKNSH with the same ED_50_ of 0.2 µg/mL. It is noteworthy that diosquinone has an excellent cytotoxicity capability against vinblastine or multiple-drug-resistant human nasopharyngeal cancer (KB-V(V-VLB)), with an ED_50_ range of 1–1.7 µg/mL.

Aderbauer et al. [67] investigated the cytotoxicity of *D. mespiliformis* dichloromethane leaf extracts against fibroblast-like mammalian cells. The findings were summarized as the minimum toxic concentration (MTC), which is the concentration at which fibroblast damage in the form of morphological change or ablation could be seen under a microscope. The leaf extract was not toxic at the tested MTC values of more than 500 µg/mL. *D. mespiliformis* 70% ethanol leaf extract was tested for in vitro cytotoxicity against MRC-5 fibroblasts and showed a modest toxicity, with an IC_50_ of >64 µg/mL [69]. Adoum [79] investigated the cytotoxic effects of *D. mespiliformis* ethanol extracts, which were reported as IC_50_ values in μg/mL. Extracts prepared from the plant were solvent partitioned and screened for activity in the brine shrimp (*Artemia cysts*) lethality test (BST). *D. mespiliformis* showed a very low brine shrimp lethality at LC_50_ > 1000 μg/mL. By growing rat skeletal myoblast L6 cells in the presence of *D. mespiliformis* dichloromethane (DCM) and 50% MeOH roots extracts, spanning a concentration range of 0.002 to 100 µg/mL, Bapela et al. [76] evaluated the in vitro inhibition of mammalian cell proliferation. The DCM extract showed a level of toxicity against the test cell line, with an IC_50_ of 24.3 µg/mL and 60.4 µg/mL in the 50% MeOH extract. The toxic properties of several root bark extracts of *D. mespiliformis* were examined by Mustapha et al. [9] utilizing brine shrimp cytotoxicity. The lethality test of the *D. mespiliformis* root bark extracts was evaluated using brine shrimp (*Artemia salina*) nauplii as the test organism. The *n*-hexane extract had the highest lethal dose concentration of 8203.52 μg/mL compared to the water extract, which was 100% safe at the investigated concentrations.

#### 2.5.9. In Vivo Studies

The neuropharmacological effects of the aqueous extract of *D. mespiliformis* stem bark were examined in mice by Adzu et al. [80]. The extracts (100 and 200 mg/kg p.o.) significantly (PB/0.05) increased the duration of pentobarbital-induced sleep and decreased exploratory and spontaneous motor behavior. However, the extract barely protected mice from death brought on by pentylenetetrazole, and only protected against the commencement of stages of seizure activity. Additionally, it had no impact on the motor coordination test. The effects of sub-chronic treatment with crude *D. mespiliformis* root extracts on a few biochemical markers in mice were examined by Jigam et al. [79]. There was minimal variation in the packed cell volumes or overall body weights of the animals given the extracts. In relation to some organ weights, triacyglycerides (148.25 ± 2.78 mg/dL) and alkaline phosphatase (41.50 ± 1.71 mg/dL) were not statistically significant (*p* > 0.05). However, there were significant (*p* > 0.05) differences between the animals treated with the extracts and controls in the heart (0.74%), the lungs (4.43%), glucose (113.92 ± 2.43 mg/dL), the total proteins (4.75 ± 1.25 mg/dL), aspartate transaminase (40.50 ± 1.50 L), and alanine transaminase (43.52 4.50 L). The effects of the acute and chronic toxicity profile of ethanolic root extracts on the clinical, hematological, and biochemical parameters of albino rats were studied by Luka et al. [25]. The intraperitoneal LD_50_ of the extract was 570 mg/kg. Although there was no statistically significant change in body weight (*p* > 0.05), there was a substantial increase in hematological parameters such as the packed cell volume, the hemoglobin concentration, red blood cells, white blood cells, and differential leucocyte counts following delivery (*p* > 0.05). The mean corpuscular volume, mean corpuscular hemoglobin, and mean corpuscular hemoglobin concentrations all increased significantly (*p* > 40.05) in a comparable manner. The acute and subchronic toxicity of the crude methanolic extract of *D. mespiliformis* and its fraction (hexane, ethyl acetate, and butanol) in Wistar rats was evaluated by Ebbo et al. [88]. Acute oral administration of the methanolic extract (5 g/kg bw) did not result in death, overt behavioral alterations, or any other physiological activities, and the LD_50_ of the crude methanolic leaf and bark extract was higher than 5 g/kg bw in Wistar rats. In a 28-day repeated dosage oral toxicity experiment, no notable adverse effects were observed in any of the parameters studied.

To evaluate the gastroprotective efficacy of leaf aqueous extracts of *D. mespiliformis*, Amang et al. [75] employed three experimental models of stomach ulcers in mice: HCl/ethanol, HCl/ethanol with indomethacin pre-treatment, and indomethacin (p.o). By administering the extract, stomach lesions caused by necrotizing drugs were avoided. The extract reduced the risk of developing ulcers after HCl/ethanol induction by 28.36%, 29.19%, and 35.82% at doses of 100, 200, and 400 mg/kg, respectively. Indomethacin pre-treatment reduced the extract’s preventive effectiveness to 19.69% and 28.24% at 200 and 400 mg/kg, respectively. During indomethacin induction, the extract at 200 mg/kg had the highest level of ulcer inhibition (88.13%). For each of the three induction models, a significant increase in mucus secretion between 44.75% and 121.34% was observed. Nwaogu et al. [82] reported the findings of acute administration of *D. mespiliformis* stem bark extracts in methanol at a dose of 5000 mg/kg body weight. After 48 h of observation, an acute dose of 5000 mg/kg body weight of methanol stem bark extract did not cause any deaths. Therefore, it was concluded that the extract’s median lethal dosage (LD_50_) was greater than 5000 mg/kg body weight. The extracts had no noticeable negative behavioral effects, such as retching, depression, tremors, weakness, refusing food and water, salivation, discharge from the eyes and ears, skin changes, or hair loss. Vandi et al. [8] detailed the antisecretory mechanism of *D. mespiliformis* root bark aqueous extract in Wistar rats. Three experimental animal models of excessive stomach acid secretion were used to test the extract: pyloric ligation, pyloric ligation plus histamine, and carbachol pretreatments. The ulcerated surface, the amount of mucus, the pH, the gastric acidity, and the pepsin activity were all measured. Malondialdehyde, superoxide dismutase, catalase, and reduced glutathione have all been identified as in vivo indicators of oxidative stress. Root bark aqueous extracts increased the mucus mass and stomach ulcer inhibition percentages in the three models under study, ranging from 9.50% to 59.52%. This increase was accompanied by a reduction in acidity and pepsin activity. Administration of the root bark aqueous extract of *D. mespiliformis* resulted in a significant decrease (*p* < 0.05, *p* < 0.01) in malondialdehyde levels, correlated with a significant increase (*p*< 0.05, *p* < 0.01) in catalase and nitrite levels compared with the negative control.

## 3. Materials and Methods

From multiple databases, including Science Direct, Google Scholar, Scopus, Web of Science, and Pub Med, every relevant scientific paper on the botanical description, traditional medicinal uses, phytochemical constituents, pharmacological and biological activities, clinical studies, and toxicology of *Diospyros mespiliformis* was retrieved. During the literature search, search terms including “*Diospyros mespiliformis*”, “traditional use”, “ethnomedicinal use”, “biological activity”, “toxicity”, “phytochemistry”, and “isolated compound” were combined.

## 4. Conclusions and Future Perspectives

*D. mespiliformis* has been used traditionally to treat a wide range of bacterial, fungal, parasitic, and viral diseases mainly in southern and western Africa. This plant has been shown in numerous studies to have strong pharmacological efficacy against a range of bacteria, fungi, viruses, and parasites. This review reports the first comprehensive summary of the traditional uses, phytochemical constituents, pharmacological activities, toxicity, and some in vivo studies of *D. mespiliformis*. The plant has a wide range of traditional uses, bioactive compounds, and pharmacological activities. According to the literature review, it can serve as a potential source of antimicrobial, antiparasitic, antiviral, anti-inflammatory, hypoglycemic, and antioxidant activities. Some of the different parts of the plant have also been tested for in vivo pharmacological activity and toxicity. The pharmacological effects of several compounds characterized by *D. mespiliformis* have still not been investigated. Therefore, in vitro and in vivo investigations to determine the potential efficacy and toxicity profile of these isolated compounds could fill in the identified gaps. No molecular research has been performed on this plant. Consequently, further research is needed to comprehend the molecular mechanisms underlying the documented pharmacological effects of the extracts and identified isolated compounds against a range of infectious illnesses. It is crucial to note that most of the data that were reviewed were evaluations of in vitro pharmacological activities. Considering the widespread antibiotic resistance, more work is required to gather more thorough evidence that will validate the in vitro findings in in vivo animal models. Furthermore, extensive pre-clinical and clinical research is also needed to determine the efficacy of this plant to establish it and its constituents as a potential effective alternative for disease prevention.

## Figures and Tables

**Figure 1 molecules-28-07759-f001:**
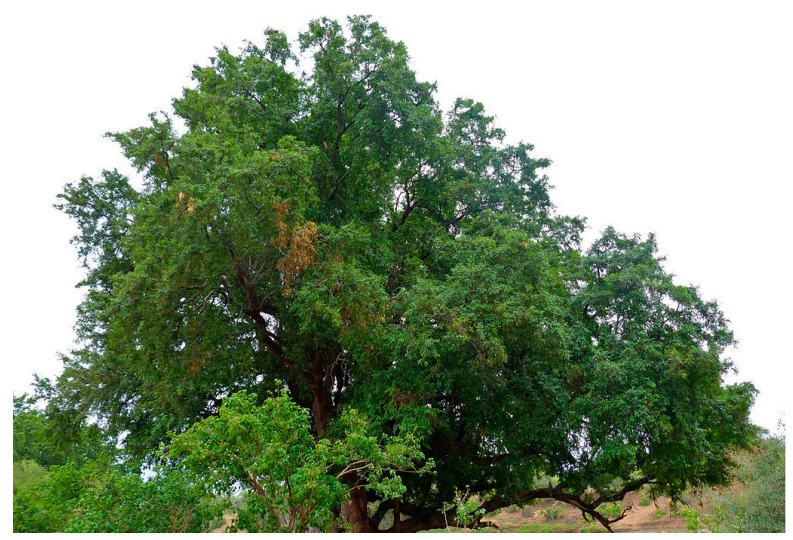
*D. mespiliformis* tree; https://www.getaway.co.za/environment/four-tree-species-have-been-added-to-south-africas-national-forest-act/ (accessed on 17 October 2023).

**Figure 2 molecules-28-07759-f002:**
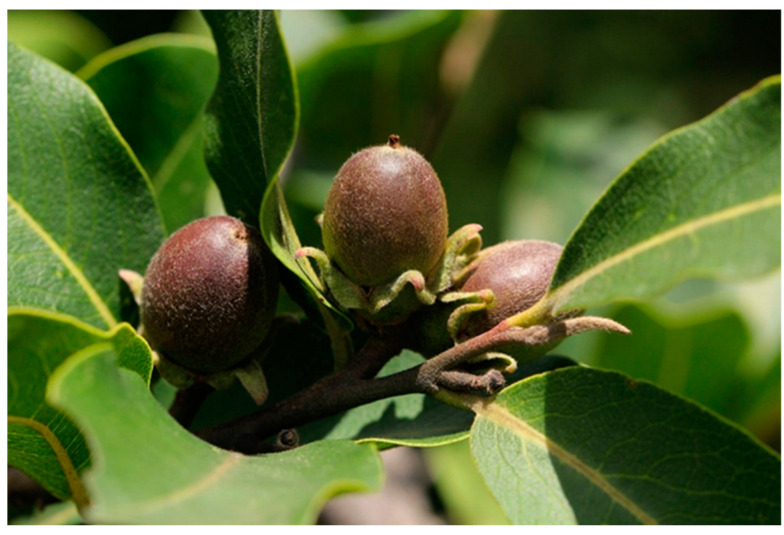
*D. mespiliformis* leaves and fruits; https://suntrees.co.za/diospyros-mespiliformis-jackalberry-jakkalsbessie-motlouma/ (accessed on 31 July 2023).

**Figure 3 molecules-28-07759-f003:**
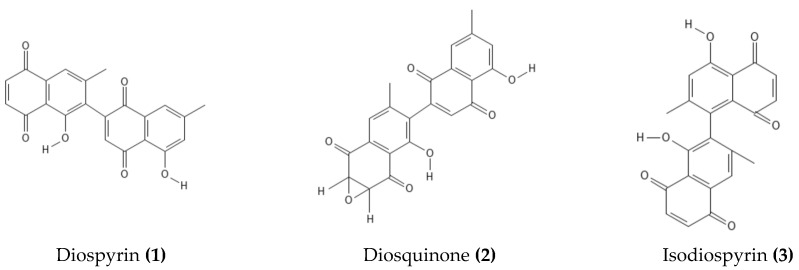
Major phenolic compounds in *D. mespiliformis*.

**Figure 4 molecules-28-07759-f004:**
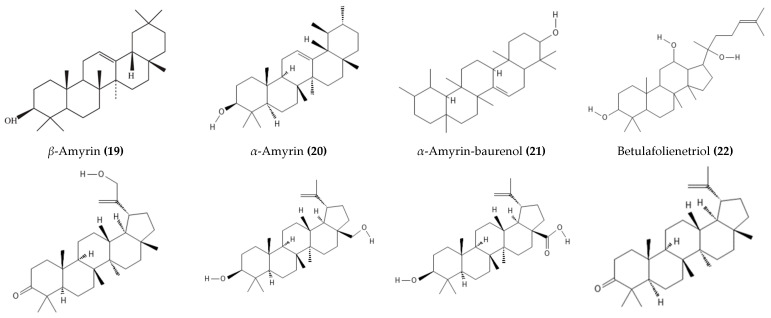
Triterpenoids isolated or detected from *D. mespiliformis*.

**Figure 5 molecules-28-07759-f005:**
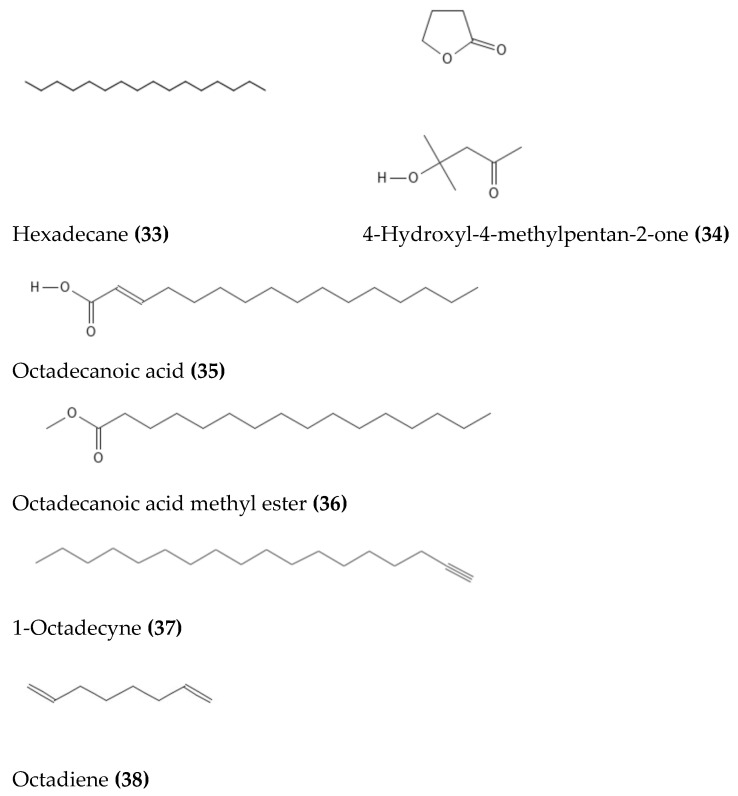
Structures of long-chain fatty acids and other classes of phytochemicals from *D. mespiliformis*.

**Table 1 molecules-28-07759-t001:** Traditional uses of the different parts of *D. mespiliformis*.

Part Used	Traditional Uses	Country	References
Leaves	Ringworm, urinary, and sexually transmitted infections, sleeping sickness, malaria, headaches, anthelmintic, wounds, dysentery, fever, leprosy, scars, skin rashes, bruises, styptic to staunch bleeding, diarrhoea, tonic, febrifuge, stomach aches, and coughs	South Africa, Ivory Coast, Nigeria, Zambia, Burkina Faso, Namibia	[15,16,17,28,30,31,32,33,34,35,36,37,38,39,40,41,42,43]
Stem	Blackleg disease in cattle, diabetes mellitus, stroke, traumatic brain injury, and malaria	South Africa, Zimbabwe, Burkina Faso, Togo	[7,26,34,44]
Bark	Oral diseases, stomach problems, diarrhoea, coughs, leprosy, STIs and urinary tract infections, dysentery, fever, vomiting, pneumonia, syphilis, and hemorrhages. Ethnoveterinary: Helminthiasis, milk production in animals, mental illness, headaches, epilepsy, and convulsions	South Africa, Burkina Faso, Nigeria, Tanzania, Senegal, Ivory Coast, Ghana, Benin, Cameroon	[18,19,23,39,40,41,42,43,45,46,47]
Roots	Ringworm, urinary, and sexually transmitted infections, abdominal pains, stomach aches, tuberculosis, male sexual dysfunction, scars, skin rashes, bruises, wounds, ringworm, dysentery, fever, coughs, epilepsy, pneumonia, syphilis, mental illness, headaches, epilepsy, convulsions, and worm expellant	South Africa, Namibia, Zimbabwe, Nigeria, Ghana, Kenya	[15,16,17,24,33,35,36,38,40,45,48,49,50,51,52]
Fruits	Dysentery, fungal infections, diarrhoea, tonic, febrifuge, skin diseases, menstrual pain, and ringworms	South Africa, Benin, Burkina Faso	[21,27,28,31,34,40]
Twigs	Teeth cleaning	South Africa	[31]

**Table 3 molecules-28-07759-t003:** Pharmacological activities of different parts of plants and major compounds of *D. mespiliformis*.

Plant Part/Compounds	Solvents Used	Pharmacological Activity	Bioassay Model	Results	References
Leaves	Acetone	Antioxidant	DPPH	IC_50_ = 25 ± 2 μg/mL	[55]
		Antibacterial	MIC (*B. stearothermophilus*)	80 µg/mL	[65]
		Antifungal	MIC (*C. albicans* and *M. canis* and *T. rubrum*)	80 µg/mL for *C. albicans*, 20 µg/mL for *M. canis* and 20 µg/mL for *T. rubrum*	[66]
	DCM	Antiproliferative	In vitro cytotoxicity	MTC > 500 µg/mL on fibroblast-like mammalian cells	[67]
	DCM: MeOH	Antibacterial	MIC (*P. acnes* ATCC 11827 and *T. mentagrophytes*)	50 µg/mL for *P. acnes* and 100 µg/mL for *T. mentagrophytes*.	[15]
		Antiparasitic	Long-term viability assay (*T. brucei*)	MIC = 500 µg/mL	[67]
	70% Ethanol	Antimicrobial	MIC (*C. albicans* ATCC 10231, *G. vaginalis* ATCC 14018, *N. gonorrhoeae* ATCC 19424 and *O. ureolytica* ATCC 43534)	3.1–6.3 mg/mL	[33]
		Antiviral	HIV-1 RT colorimetric ELISA kit (Roche)	78.7% at 0.1 mg/mL had	[68]
		Antiparasitic	In vitro antiplasmodial activity	IC_50_ = 25.8 µg/mL for *Trypanosoma cruzi*, IC_50_ = >64 µg/mL for *Leishmania infantum*	[69]
		Antiproliferative	In vitro cytotoxicity	IC_50_ = >64 µg/mL for MRC-5 fibroblasts	[69]
	Ethanol	Antiviral	In vitro allantoic sac routes of developing chick embryos	95.0%, 90.5%, and 89.0% at 400 mg/mL, 200 mg/mL, and 100 mg/mL respectively, for Newcastle disease virus	[70]
		Anti-hypersensitivity	Intracellular free calcium measurements	Reduced amplitude of Ca^2+^ release from SR at 10 mg/mL. IC_50_ = 9.23 mg/mL and 54% inhibited calcium release.	[71]
	MeOH	Antioxidant	DPPH	IC_50_ = 6.94 ± 0.49 µg/mL	[72]
		Antimycobacterial	MIC (*M. smegmatis*)	167 µg/mL	[5]
		Antiparasitic	In vitro antiplasmodial bioassay	IC_50_ = 1.51 µg/mL for *P. falciparum* 3D7A	[73]
		Antiviral	In vitro allantoic sac routes of developing chick embryos	100.0%, 92.8%, and 90.5% at 400 mg/mL for Newcastle disease virus	[70]
		Toxicity	Acute and subchronic toxicity in rats	LD_50_ of >5g/kg. No notable adverse effects seen on parameters studied	[72]
	Water	Antioxidant	ABTS, FRAP	1.17 ± 0.00 TEAC in mM (ABTS). 70.77 ± 0.4 M ET/g	[74]
		Antifungal	MIC (*C. albicans*, and *T. rubrum*)	20 µg/mL for *C. albicans*, 40 µg/mL for *T. rubrum*.	[66]
		Antiparasitic	In vitro antiplasmodial bioassay (*P. falciparum* 3D7A)	IC_50_ = 3.01 µg/mL	[73]
		Antiviral	In vitro allantoic sac routes of developing chick embryos	91.0%, 86.0%, and 85.0% at 400 mg/mL, 200 mg/mL, and 100 mg/mL, respectively, for Newcastle disease virus	[70]
		Anti-hypersensitivity	Intracellular free calcium measurements	IC_50_ = 8.84 mg/mL at 10 mg/mL. 29% inhibited calcium release.	[71]
		Toxicity	Gastroprotective efficacy: stomach ulcer	200 mg/kg had the highest level of ulcer inhibition (88.13%)	[75]
Leaf fractions	Butanol	Antioxidant	DPPH	IC_50_ = 1.44 ± 0.01 µg/mL	[55]
	Hexane	Antioxidant	DPPH	IC_50_ = 28.03 ± 2.57 µg/mL	[55]
	Ethyl acetate	Antioxidant	DPPH	IC_50_ = 1.08 ± 0.04 µg/mL	[55]
	Water	Antioxidant	DPPH	IC_50_ = 4.73 ± 0.23 µg/mL	[55]
Roots	DCM: 50% MeOH	Antiparasitic	In vitro hypoxanthine incorporation assay (*P. falciparum* NF54)	IC_50_ = 4.40/28.4 µg/mL	[20]
			Antileishmanial, resazurin assay (*L. donovani* MHOM-ET-67/L82)	IC_50_ = 7.7 µg/mL for DCM and IC_50_ = 54 µg/mL for 50% MeOH.	[76]
		Antiproliferative	In vitro inhibition of mammalian cell proliferation	IC_50_ = 24.3 µg/mL for DCM and 60.4 µg/mL for MeOH	[77]
	Ethanol	Antiparasitic	Acute toxicity and prolonged administration in rats	Intraperitoneal LD_50_ of 570 mg/kg	[25]
	70% Ethanol	Anti-inflammatory	15-LOX	IC_50_ = 188.1 µg/mL	[33]
		Antiviral	HIV-1 RT colorimetric ELISA kit (Roche)	17.4% inhibition	[33]
	MeOH	Antioxidant	DPPH	IC_50_ = 3.47 ± 0.05 µg/mL	[55]
		Antiparasitic	In vitro antiplasmodial bioassay (*P. falciparum* 3D7A)	IC_50_ = 2.12 µg/mL	[73]
			In vivo antiplasmodial activity in mice (*Plasmodium berghei*)	High rate of parasite clearance (84.7%) and lower parasitemia (0.67%)	[78]
		Toxicity	Subchronic in vivo studies	Safe dose of 400mg/Kg bw and LD_50_ of 620mg/kg bw of mice.	[79]
	Water	Antibacterial	Disc diffusion (*S. aureus, P. aeruginosa, E. coli* and *Shigella* spp).	10–13 mm on *S. aureus*, 11–13 mm on *P. aeruginosa*, 11–14 mm on *E. coli* and 10–11 mm on *Shigella* spp.	[4]
		Antiparasitic	In vitro antiplasmodial bioassay (*P. falciparum* 3D7A)	IC_50_ = 2.91 µg/mL	[73]
Root bark	Acetone	Anti-inflammatory	XO, NO	IC_50_ = 142 8 µg/mL (XO) and IC_50_ = 79.8 ± 2.7 µg/mL (NO)	[80]
	Hexane	Antiproliferative	Brine shrimp (*Artemia salina*) cytotoxicity	8203.52 μg/mL lethal dose	[9]
	Water	Antiproliferative	Brine shrimp (*Artemia salina*) cytotoxicity	100% safe at 10–1000 μg/mL	[9]
		Antioxidant	ABTS, DPPH and FRAP	IC_50_ = 220 µg/mL for ABTS, 494 µg/mL DPPH and 543 µg/mL	[8]
		Antisecretory mechanism	Pyloric ligation, pyloric ligation plus histamine, and carbachol pretreatments	Increased mucus mass and stomach ulcer inhibition ranging from 9.50% to 59.52%	[8]
Bark	95% Ethanol	Antiparasitic	In vivo antitrypanosomal activity of *Trypanosoma evansi*-infected rats	Increased red blood cells and elevated bilirubin. Reduced total proteins	[77]
	Hexane	Antimycobacterial	MIC (*M. tuberculosis H_37_R_a_*)	100 µg/mL	[50]
	MeOH	Antioxidant	DPPH	IC_50_ = 7.82 ± 0.76 µg/mL	[55]
		Antiparasitic	In vivo antiplasmodial activity in mice (*Plasmodium berghei* NK65)	53% at 800 mg/kg dosage	[81]
Bark fractions	Ethyl acetate and hexane	Anti-inflammatory	Wound healing	Fully healed	[55]
Stem	Ethanol	Antiproliferative	Brine shrimp (*Artemia cysts*) lethality test (BST)	LC_50_ >100 μg/mL	[79]
Stem bark	MeOH	Antipyretic	In vivo studies	LD_50_ = 513.80 ± 33.92 mg/kg i.p. in mice.	[63]
		Antiparasitic	In vivo antiplasmodial activity against *P. berghei* ANKA in mice	Parasitemia (5 ± 1), increased packed cell volume (36% ± 1.4), increased platelets (2 ± 1.4 105 mm^3^), decreased alkaline phosphatase (56 ± 0.7 U/L), alanine aminotransferases (6.2 ± 0.8 U/L), and alanine aminotransferases (8 ± 3.8 U/L).	[12]
		Toxicity	Acute toxicity and hepatoprotective effects	LD_50_ > 5000 mg/kg bw. Possess hepatoprotective property by inhibiting lipid peroxidation.	[82]
	Water	Neuropharmacological	In vivo studies in mice	Increased pentobarbital-induced sleep, decreased exploratory and spontaneous motor behavior	[83]
Stem fractions	DCM	Antiparasitic	In vivo antiplasmodial activity against *P. berghei* NK 65 in mice	High parasite clearance	[11]
Fruits	Ethanol	Antioxidant	DPPH. In vivo antioxidants in rats.	IC_50_ = 1.037 ± 0.204 mg/mL. Increased the levels of the enzymes SOD, catalase, peroxidase, alanine transaminase, and aspartate aminotransferase	[84]
	Hydroethanolic	Antioxidant	DPPH	IC_50_ = 1.111 ± 0.135 mg/mL	[84]
	MeOH	Antioxidant	DPPH radical scavenging, reducing power effects, and superoxide-anion-radical scavenging.	Increase radical-scavenging effect, reducing power and superoxide-anion-radical-scavenging.	[85]
		Antioxidant	DPPH, H_2_O_2_ scavenging	87.36% at 1 mg/mL for DPPH. >85% at 1 mg/m for H_2_O_2_.	[58]
Isoscutellarein 7-*O*-(4′′′-*O*-acetyl)-*β*-allopyranosyl (1′′′ → 2″)-*β*-glucopyranoside (**8**)		Antioxidant	DPPH	IC_50_ = 15.46 μg/mL	[65]
Luteolin 3′,4′,6,8-tetramethyl ether (**9**)		Antimycobacterial	MIC (*M. smegmatis*)	250 µg/mL	[5]
		Antibacterial	Disc diffusion (*E. coli*)	34 mm	[65]
		Antibacterial	MIC (*S. aureus*)	9.77 μg/mL	[65]
		Antioxidant	DPPH	IC_50_ = 15.46 μg/mL	[65]
Quercetin 3-*O*-*α*-rhamnoside (**15**)		Antibacterial	MIC (*S. aureus* NCTC 6571, *S. aureus* E3T, *E. coli* KL16 and *P. aeruginosa* NCTC 6750)	3 to 30 μg/mL for *S. aureus*, 15 for *E. coli* and 16 μg/mL for *P. aeruginosa*	[64]
		Antioxidant	DPPH	IC_50_ = 12.32 μg/mL	[65]
Diosquinone (**2**)		Antiproliferative	In vitro cytotoxicity	ED_50_ = 0.18 μg/mL for U373 cells, ED_50_ = 0.2 µg/mL for BC-1, HT-1080, Lu-1, KB, and SKNSH cells, ED_50_ = 1–1.7 µg/mL for KB-V(V-VLB) cells	[36]
Plumbagin (**12**)		Antidiabetic	α- Glucosidase enzyme inhibition assay	IC_50_ = 0.002 ± 0.004 mM	[14]
Lupeol (**27**)		Antidiabetic	α- Glucosidase enzyme inhibition assay	IC_50_ = 0.46 ± 0.002 mM	[14]
Betulin (**24**)		Antidiabetic	α- Glucosidase enzyme inhibition assay	IC_50_ = 0.0624 ± 0.002 mM	[14]

## Data Availability

The data presented in this study are available upon request from the corresponding authors.

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
