# Peer review of "Traditional Uses, Pharmacological Activities, and Phytochemical Analysis of Diospyros mespiliformis Hochst. ex. A. DC (Ebenaceae): A Review"

_molecules, 2023, doi:10.3390/molecules28237759_

Round 1

Reviewer 1 Report

Comments and Suggestions for Authors

Comments to authors:

Abstract; Line 16, steroids/triterpenoids, both are different classes should be mentioned separately (i.e., terperoids, triterpenoids,...etc.).

Line 14-18; The authors mentioned “phytochemical screening reports” and “secondary metabolites tentatively identified”. Preliminary phytochemical screening should not be relying on, however, isolation or chromatographic and/or spectroscopic methods are only trustful. Thus, preliminary screening should be deleted from abstract but may be kept in discussion section under a separate title/sub-section.

Line 101: In Table 1. Traditional uses of the different parts of D. mespiliformis, the country of traditional use should be added to the table.

Lines 102 & 103; 2.2.1. Phytochemical analysis section, should not be a subtitle of 2.2. Pharmacological Activity section. But they should be separate parts.

Regarding the 2.2.1. Phytochemical analysis section, no sufficient effort has been made to classify the reported structures or to add subsections of each phytochemical group. The review in this section appeared too preliminary in this regard.

The chemical review section should be more organized. Thus, I suggested the following:

· The compounds should be classified into distinct classes.

· Each class should be collected in one figure and listed consequently in the table.

· The structures should be listed alphabetically.

· In the table of reported phytochemicals, the authors should include the detection/isolation method e.g., GC-MS, UPLC, isolated, ...etc.

· The table of structures, the part used i.e., leaf, fruit, bark,..etc. from which the phytochemicals have been identified.

· Accordingly, compounds should be renumbered after the new classification.

· All structures must be drawn with professional software like ChemDraw software to maintain the style of chemical drawings.

2.2. Pharmacological Activity; a table summarizing the pharmacological activities is highly recommended. This table may include biological Activity, extract/

compound, bioassay model, results, and references.

The fruits of D. mespiliformis are edible and have nutritional values. However, the authors didn’t describe their nutritional values (please see the following articles: https://pubmed.ncbi.nlm.nih.gov/16395630/  & https://pubmed.ncbi.nlm.nih.gov/30897690/ &

https://pubmed.ncbi.nlm.nih.gov/9201749/

The title clearly addressed the “Toxicology” of the plant, however, no clear section described the toxicological properties of the plant. The toxicological properties ashowed be properly discussed and reviewed or deleted from the title.

The authors should consider the important related articles:

Most important: https://pubmed.ncbi.nlm.nih.gov/35245970/

Other:

https://onlinelibrary.wiley.com/doi/abs/10.1002/ptr.2650090508

https://pubmed.ncbi.nlm.nih.gov/12413720/

https://pubmed.ncbi.nlm.nih.gov/36343283/

https://pubmed.ncbi.nlm.nih.gov/26904625/

https://pubmed.ncbi.nlm.nih.gov/12241995/

https://pubmed.ncbi.nlm.nih.gov/12672163/

Comments on the Quality of English Language

A major revision is required.

Author Response

Comments and Suggestions for Authors

Comments to authors: Reviewer 1

Comment 1: Abstract; Line 16, steroids/triterpenoids, both are different classes should be mentioned separately (i.e., terpenoids, triterpenoids,...etc.).

Response: The two classes have been separated. Thank you.

Comment 2: Line 14-18; The authors mentioned “phytochemical screening reports” and “secondary metabolites tentatively identified”. Preliminary phytochemical screening should not be relying on, however, isolation or chromatographic and/or spectroscopic methods are only trustful. Thus, preliminary screening should be deleted from abstract but may be kept in discussion section under a separate title/sub-section.

Response: We have deleted the sentence from the abstract as per the recommendation. Thank you.

Comment 3: Line 101: In Table 1. Traditional uses of the different parts of D. mespiliformis, the country of traditional use should be added to the table.

Response: The column has been added to the table. Thank you.

Comment 4: Lines 102 & 103; 2.2.1. Phytochemical analysis section, should not be a subtitle of 2.2. Pharmacological Activity section. But they should be separate parts.

Response: The two sections have been separated. Thank you

Comment 5: Regarding the 2.2.1. Phytochemical analysis section, no sufficient effort has been made to classify the reported structures or to add subsections of each phytochemical group. The review in this section appeared too preliminary in this regard.

Response: Thank you for this comment, we have added some information. The information reviewed in this section is predicated on the data that are currently available, since most studies on the different parts of D. mespiliformis have mostly examined the presence of primary metabolites and secondary metabolites group, and not necessarily identifying them. 

The chemical review section should be more organized. Thus, I suggested the following:

  • The compounds should be classified into distinct classes.

Response: We have classified the compounds into classes were possible. Thank you.

  • Each class should be collected in one figure and listed consequently in the table.

Response: Done. Thank you.

  • The structures should be listed alphabetically.

Response: The structures were listed alphabetically in each class.

  • In the table of reported phytochemicals, the authors should include the detection/isolation method e.g., GC-MS, UPLC, isolated, ...etc.

Response: A column was added on the table to include detection/isolation methods.

  • The table of structures, the part used i.e., leaf, fruit, bark, ..etc. from which the phytochemicals have been identified.

Response: Done, thank you.

  • Accordingly, compounds should be renumbered after the new classification.

Response: Done, thank you.

  • All structures must be drawn with professional software like ChemDraw software to maintain the style of chemical drawings.

 Response: We have redrawn the structures as per the recommendation.

Comment 6: 2.2. Pharmacological activity; a table summarizing the pharmacological activities is highly recommended. This table may include biological Activity, extract/compound, bioassay model, results, and references.

Response: Done

Comment 7:

https://pubmed.ncbi.nlm.nih.gov/16395630/ Glew, R.S., Vanderjagt, D.J., Chuang, L.T., Huang, Y.S., Millson, M. and Glew, R.H., 2005. Nutrient content of four edible wild plants from West Africa. Plant Foods for Human Nutrition60, pp.187-193. &

https://pubmed.ncbi.nlm.nih.gov/30897690/ Achaglinkame, M.A., Aderibigbe, R.O., Hensel, O., Sturm, B. and Korese, J.K., 2019. Nutritional characteristics of four underutilized edible wild fruits of dietary interest in Ghana. Foods8(3), p.104. &

https://pubmed.ncbi.nlm.nih.gov/9201749/ Petzke, K.J., Ezeagu, I.E., Proll, J., Akinsoyinu, A.O. and Metges, C.C., 1997. Amino acid composition, available lysine content and in vitro protein digestibility of selected tropical crop seeds. Plant Foods for Human Nutrition50, pp.151-162.

Response: The information has been added under the phytochemical section. Thank you.

Comment 8: The title clearly addressed the “Toxicology” of the plant, however, no clear section described the toxicological properties of the plant. The toxicological properties ashowed be properly discussed and reviewed or deleted from the title.

Response: These constructive corrections are truly welcomed. Accordingly, we have deleted the toxicology from the heading.

Comment 9: The authors should consider the important related articles:

Most important: https://pubmed.ncbi.nlm.nih.gov/35245970/ Hawas, U.W., El-Ansari, M.A. and El-Hagrassi, A.M., 2022. A new acylated flavone glycoside, in vitro antioxidant and antimicrobial activities from Saudi Diospyros mespiliformis Hochst. ex A. DC (Ebenaceae) leaves. Zeitschrift für Naturforschung C77(9-10), pp.387-393.

Other:

https://onlinelibrary.wiley.com/doi/abs/10.1002/ptr.2650090508 Lajubutu, B.A., Pinney, R.J., Roberts, M.F., Odelola, H.A. and Oso, B.A., 1995. Antibacterial activity of diosquinone and plumbagin from the root of Diospyros mespiliformis (Hostch)(Ebenaceae). Phytotherapy Research9(5), pp.346-350.

https://pubmed.ncbi.nlm.nih.gov/12413720/ Adzu, B., Amos, S., Muazzam, I., Inyang, U.S. and Gamaniel, K.S., 2002. Neuropharmacological screening of Diospyros mespiliformis in mice. Journal of ethnopharmacology83(1-2), pp.139-143.

https://pubmed.ncbi.nlm.nih.gov/36343283/ Olanlokun, J.O., Adetutu, J.A. and Olorunsogo, O.O., 2021. ln vitro inhibition of beta-hematin formation and in vivo effects of Diospyros mespiliformis and Mondia whitei methanol extracts on chloroquine-susceptible Plasmodium berghei-induced malaria in mice. Interventional Medicine and Applied Science11(4), pp.197-206.

https://pubmed.ncbi.nlm.nih.gov/26904625/ Adewuyi, A. and Oderinde, R.A., 2014. Fatty acid composition and lipid profile of Diospyros mespiliformis, Albizia lebbeck, and Caesalpinia pulcherrima seed oils from Nigeria. International journal of food science2014.

https://pubmed.ncbi.nlm.nih.gov/12241995/ Adzu, B., Amos, S., Dzarma, S., Muazzam, I. and Gamaniel, K.S., 2002. Pharmacological evidence favouring the folkloric use of Diospyros mespiliformis Hochst in the relief of pain and fever. Journal of Ethnopharmacology82(2-3), pp.191-195.

https://pubmed.ncbi.nlm.nih.gov/12672163/ Adeniyi, B.A., Robert, M.F., Chai, H. and Fong, H.H.S., 2003. In vitro cytotoxicity activity of diosquinone, a naphthoquinone epoxide. Phytotherapy Research17(3), pp.282-284.

Response: The information of most of the articles mentioned was already included. We have incorporated the information of the few articles which were not included. Thank you so much for this comment.

Reviewer 2 Report

Comments and Suggestions for Authors

The subject of the manuscript is an interesting one, and the review is quite complete and well structured. But for its publication some additions are necessary:

- structuring the information of extracts of Diospyros mespiliformis that have demonstrated remarkable efficiency for medicinal use in a table, specifying the type of study, administered dose, therapeutic and adverse effects, bioactive compounds

- a more detailed explanation of the aspects related to phytotoxicity 

- discussions of possible interactions of the active principles with different food products and nutritional supplements

- a discussion on restrictions on therapeutic recommendations

- the specification of future perspectives in terms of therapeutic valorization 

Author Response

Comments and Suggestions for Authors

Comments to authors: Reviewer 2             

The subject of the manuscript is an interesting one, and the review is quite complete and well structured. But for its publication some additions are necessary:

Comment 1: - structuring the information of extracts of Diospyros mespiliformis that have demonstrated remarkable efficiency for medicinal use in a table, specifying the type of study, administered dose, therapeutic and adverse effects, bioactive compounds.

Response: Two tables were added to address the above comment.

Comment 2: - a more detailed explanation of the aspects related to phytotoxicity. 

Response: The word toxicology has deleted from the title inline with recommendation by the other reviewer as well.

Comment 3: - discussions of possible interactions of the active principles with different food products and nutritional supplements

Response: Thank you for this, we have added some more information under the phytochemical analysis section.

Comment 4: - a discussion on restrictions on therapeutic recommendations

Response: An information has been added under conclusion and future perspectives which addressed this comment

Comment 5: - the specification of future perspectives in terms of therapeutic valorization 

Response: This has been added under the conclusion section.

Round 2

Reviewer 1 Report

Comments and Suggestions for Authors

Comments to authors:

Line 135: cysteine and methionine (95%),...change to  Cysteine...

Line 139: "Rimelia reticulata" in italic font ...change to "Rimelia reticulata" in italic font.

Lines 103-156, please add a subtitle 2.2.1. Nutritional values (or primary metabolites) of D. mespiliformis.

move it down to line 261, directly before talking about different pharmacological activities.

Lines 157-260: The authors started talking about secondary metabolites. So, a new subtitle "2.2.2. Secondary metabolites of D. mespiliformis” should be added to this section.

Line 167: 2.3. Pharmacological Activity-subtitle- should be moved down to start from line 261, directly before talking about different pharmacological activities. After that, renumbering of subtitles 2.3.1 Antimicrobial Activity, 2.3.2.  Anti-inflammatory Activity, 2.3.3. Antiparasitic Activity, 2.3.4. Antidiabetic Activity, 2.3.5. Antiviral Activity, 2.3.6. Anti-hypersensitivity Activity, 2.3.7. Antioxidant Activity, 2.3.8. Antiproliferative activity, and 2.3.9. In vivo studies.

Lines 194-201: Use bold font for newly added compounds' numbering. Also, use italic font for atoms of linkages and Latin symbols.

Table 2. use italic font for atoms of linkages and Latin symbols. Also, use uppercase letters to start names in second column.

Page 11-12: Figure 4. can be in one page by decreasing the structures’ sizes and spaces between them. Please apply this for other structures’ figures too (see attached pdf).

Line 305: Please use proper citation and correct the reference  ..[63.

For other comments, please see the attached pdf.

Comments on the Quality of English Language

Minor English language revision is needed.

Author Response

Molecules-2680904

Title: Traditional uses, pharmacological activities, toxicology, and phytochemical analysis of Diospyros mespiliformis Hochsr. Ex. A. DC (Ebenaceae): A review

Author's Reply to the Review Report (Reviewer 1)

Comments and Suggestions for Authors

Comments to authors:

Comment: Line 135: cysteine and methionine (95%),...change to  Cysteine...

Response: Done

Comment: Line 139: "Rimelia reticulata" in italic font ...change to "Rimelia reticulata" in italic font.

Response: Done

Comment: Lines 103-156, please add a subtitle 2.2.1. Nutritional values (or primary metabolites) of D. mespiliformis.

Response: Done

Comment: move it down to line 261, directly before talking about different pharmacological activities.

Response: Done

Comment: Lines 157-260: The authors started talking about secondary metabolites. So, a new subtitle "2.2.2. Secondary metabolites of D. mespiliformis” should be added to this section.

Response: Done

Comment: Line 167: 2.3. Pharmacological Activity-subtitle- should be moved down to start from line 261, directly before talking about different pharmacological activities. After that, renumbering of subtitles 2.3.1 Antimicrobial Activity, 2.3.2.  Anti-inflammatory Activity, 2.3.3. Antiparasitic Activity, 2.3.4. Antidiabetic Activity, 2.3.5. Antiviral Activity, 2.3.6. Anti-hypersensitivity Activity, 2.3.7. Antioxidant Activity, 2.3.8. Antiproliferative activity, and 2.3.9. In vivo studies.

Response: The subheading has been moved and the numbering has been modified accordingly.

Comment: Lines 194-201: Use bold font for newly added compounds' numbering. Also, use italic font for atoms of linkages and Latin symbols.

Response: Done

Comment: Table 2. use italic font for atoms of linkages and Latin symbols. Also, use uppercase letters to start names in second column.

Response: Done

Comment: Page 11-12: Figure 4. can be in one page by decreasing the structures’ sizes and spaces between them. Please apply this for other structures’ figures too (see attached pdf).

Response: The sizes of all the structures have been reduced.

Comment: Line 305: Please use proper citation and correct the reference  ..[63.

Response: Done

Reviewer 2 Report

Comments and Suggestions for Authors

Accept the manuscript in the revised form

Author Response

N/A